# 3D-Printed Microfluidic Chip for Real-Time Glucose Monitoring in Liquid Analytes

**DOI:** 10.3390/mi14030503

**Published:** 2023-02-21

**Authors:** Ivana Podunavac, Miroslav Djocos, Marija Vejin, Slobodan Birgermajer, Zoran Pavlovic, Sanja Kojic, Bojan Petrovic, Vasa Radonic

**Affiliations:** 1University of Novi Sad, BioSense Institute, Dr Zorana Đinđića 1, 21000 Novi Sad, Serbia; 2University of Novi Sad, Faculty of Technical Sciences, Trg Dositeja Obradovića 6, 21000 Novi Sad, Serbia; 3University of Novi Sad, Faculty of Medicine, Hajduk Veljkova 3, 21000 Novi Sad, Serbia

**Keywords:** microfluidics, lab-on-a-chip, 3D printing, SLA, PMMA, glucose, electrochemical sensor

## Abstract

The connection of macrosystems with microsystems for in-line measurements is important in different biotechnological processes as it enables precise and accurate monitoring of process parameters at a small scale, which can provide valuable insights into the process, and ultimately lead to improved process control and optimization. Additionally, it allows continuous monitoring without the need for manual sampling and analysis, leading to more efficient and cost-effective production. In this paper, a 3D printed microfluidic (MF) chip for glucose (Glc) sensing in a liquid analyte is proposed. The chip made in Poly(methyl methacrylate) (PMMA) contains integrated serpentine-based micromixers realized via stereolithography with a slot for USB-like integration of commercial DropSens electrodes. After adjusting the sample’s pH in the first micromixer, small volumes of the sample and enzyme are mixed in the second micromixer and lead to a sensing chamber where the Glc concentration is measured via chronoamperometry. The sensing potential was examined for Glc concentrations in acetate buffer in the range of 0.1–100 mg/mL and afterward tested for Glc sensing in a cell culturing medium. The proposed chip showed great potential for connection with macrosystems, such as bioreactors, for direct in-line monitoring of a quality parameter in a liquid sample.

## 1. Introduction

Microfluidics deals with the control and manipulation of fluids on a sub-millimeter scale in a single or network of microchannels [1,2,3,4,5]. With the development of technology, microfluidics is becoming a part of the complex miniaturized systems called Lab-on-a-Chip (LOC) [6,7,8] that contain multiple operations integrated into one single chip, such as reagent mixing [9], particle separation [10], DNA extraction and amplification [11], detection [12], etc. Recent studies demonstrate that LOC systems had found application in various point-of-care analyses such as the detection of foodborne pathogenic bacteria in food quality control [13,14,15], environmental monitoring [16,17], and biomarker detection [18,19], or could be used for developing an organ-on-chip for in vitro analysis [20,21].

Important parts of LOC systems are Microfluidic (MF) mixers [22,23], which can be classified as active if the external energy of the actuator accelerates the mixing process, or a passive MF mixer that operates without external actuators. Recently proposed active micromixers use an acoustic actuation [24], electric field [25], magneto-hydrodynamic force [26], etc., as an external mixing actuator. Active mixers enable faster mixing and shorter mixer length than passive ones, but their integration into complex systems is a challenging task. On the other hand, passive MF mixers contain complex shapes of MF channels to stimulate the mixing process and their mixing time is greater than the mixing time of active micromixers. Different passive MF mixers have been proposed in the literature based on different geometrical shapes, including serpentine [27], meander [28], and parallelogram barriers [29], or 3D topologies such as T-mixer, Caterpillar, or Tesla-like mixers [30]. 

A number of technologies have been used for the realization of MF devices such as PolyDiMethylSiloxane (PDMS), micromolding, micromachining, 3D printing, or xurographic technique. Even though PDMS technology produces long-lasting chips that are optically transparent and biocompatible, their fabrication process is complex, time-consuming, and costly [31]. Moreover, the PDMS process requires a complex lithography process and design optimization, which requires reparation of the complete fabrication flow. Recently, polymer microfabrication has become a popular substitute for well-established silicon and glass-based microfluidics especially with the development of novel thermoplastic materials. One microfluidic technology is based on Poly(methyl methacrylate (PMMA) polymer that has excellent optical transparency and, therefore, is widely used for biomedical applications. PMMA is usually micromachined with CNC machine (computer numerical control machine) or laser, while bonding techniques require using pressure or chemical treatment [32]. On the other hand, micromolding is a technique for the fabrication of microfluidics in polymers using molds with a negative of the desired geometry. Although micromolding technology requires several steps including the mold preparation, an imprint of polymer material, curing of the polymer, separation, and bonding, it can be utilized in many biomedical applications [33,34,35]. In addition, many hybrid fabrication technologies are proposed in the literature to overcome existing drawbacks of the previously mentioned technologies and reduce the price and fabrication time. One of the hybrid technologies we recently proposed combines the laser micromachining process of a non-sintered glass-ceramic with xurographic technique, offers rapid low-cost chip production, and enables realization of complex and multilayered channel geometries [36].

One of the developing low-cost fabrication technologies is 3D printing. As a technology that can provide geometrically complex structures in a single step at the microscale while ensuring affordable and time-efficient fabrication, 3D printing has become the cutting-edge manufacturing process for MF systems [37,38]. There are numerous 3D printing methods that can be used for a variety of applications such as fused deposition modeling [39], selective laser melting [40], photopolymer inkjet printing [41], laminated object manufacturing [42], stereolithography [43], etc. Stereolithographic (SLA) printers have revolutionized many manufacturing processes with their ability to easily produce highly precise structures. In this process, a laser is used to cure and solidify layers of photopolymer resin. The procedure begins with slicing a 3D model into tiny layers, followed by projecting a laser onto a tray of photopolymer resin. The laser selectively hardens the resin, layer by layer, constructing the design. Due to its high resolution and precision, SLA is usually a preferred choice for producing, developing, and testing different channel designs and structures. Recent studies show that SLA-fabricated MF devices can be used in a wide range of applications, from cell culture [44] to chemical synthesis [45], and biosensors [46].

Glucose (Glc) sensors have developed into the most thoroughly researched biosensing devices as a result of the enormous need for glucose monitoring in biomedicine and healthcare to offer diagnostic information about diverse biological samples, including blood, urine, interstitial fluid, perspiration, breath, saliva, and ocular fluid [47,48]. Additionally, Glc monitoring is frequently used in the food sector as a tool for quality assurance and safety [49]. Cell culture is another application for Glc quantification, which offers important insights into metabolism since Glc serves as the main carbon source for the growth and productivity of the cells. In order to maintain optimal conditions for cell growth and productivity, it is important to closely monitor Glc levels in the medium. Raman spectroscopy, High performance liquid chromotagraphy (HPLC), and other expensive and time-consuming spectroscopic techniques can be used for Glc monitoring in cell culture [50]. Various other methods have been proposed so far for Glc detection, such as enzymatic assays [51], colorimetric assays [52], and biosensors [53]. Enzymatic assays use enzymes such as glucose oxidase (GOx) to convert Glc to hydrogen peroxide (H_2_O_2_), which is then detected using a colorimetric indicator. Biosensors, such as fluorescent biosensors [54] or electrochemical biosensors [55], allow precise and real-time monitoring of Glc levels, enabling scientists to make adjustments to the medium as needed to ensure optimal cell growth and productivity. The most common method for Glc sensing in MF systems is based on enzymatic reactions, where an enzyme catalyzes the oxidation of Glc to generate H_2_O_2_. This H_2_O_2_ can then be detected using various techniques, such as electrochemical or optical methods.

In this paper, we propose a 3D printed chip with integrated passive micromixers fabricated using the 3D printing technology and laser micromachining of PMMA. The fabricated MF mixers are based on the serpentine curve which leads the mixed liquids to a sensing chamber where a gold DropSens electrochemical electrode is integrated. The reaction between Glc and GOx in the mixed sample enables sensing of the Glc concentration via electrochemical measurements. The analyzed mixer is fabricated using SLA technology and its performances, together with the sensing potential, are experimentally verified. The measurement algorithm is developed for autonomous measurements in macrosystems such as bioreactors, where the proposed chip can be used for in-line monitoring of a Glc in a cell culturing media.

## 2. Theoretical and Numerical Analysis

The 3D printed MF chip contains two MF mixers with serpentine designs and a sensing chamber with an integrated commercial DropSens electrode (as shown in Figure 1). This chip can be connected to a bioreactor, from which a medium sample can be periodically taken. The first micromixer is used to mix cell medium and acetic acid to adjust the pH of the mixed solution to 5.1, as the enzyme activity is best in the pH range of 5.1 to 5.5. An analytical calculation is made to estimate the ratio of mixing volumes, or the ratio of flow rates at the two inlets, based on the pH values of the medium and acetic acid at the two inlets. After the first micromixer, the second micromixer is mixing the pH-adjusted medium with an enzyme, GOx, and directing the mixed analyte to the sensing chamber, where the planar electrochemical electrode is integrated at the bottom. In this way, the compact LOC enables liquid mixing and sensing the response at the electrode surface.

The calculation of the ratio of flow rates of medium and acetic acid is made based on the well-known equation for mixing two analytes with different pH values, and the definition of flow rate. We begin by using the equation to estimate the pH value of a mixture of two liquids with different pH values, Equation (1). Herein, ${\left[H^{+}\right]}_{1}$, ${\left[H^{+}\right]}_{2},$ and ${\left[H^{+}\right]}_{tot}$ are concentrations of hydrogen ions in cell medium, acetic acid and in the total mixed solution, respectively, while $V_{1}$, $V_{2}\;$and $V_{tot\;}$are their volumes.
(1)$${\left[H^{+}\right]}_{1}V_{1}+{\left[H^{+}\right]}_{2}V_{2}={\left[H^{+}\right]}_{tot}V_{tot}$$

Considering that the flow time is the same for both liquids, the total volume can be expressed through the ratio of flow rates of acetic acid and medium $\frac{Q_{2}}{Q_{1}}$, Equation (2).
(2)$$V_{tot}=V_{1}\left(1+\frac{Q_{2}}{Q_{1}}\right)$$

Finally, by substituting Equation (2) in Equation (1), the final calculation is presented with Equation (3):(3)$$\frac{Q_{2}}{Q_{1}}=\frac{{\left[H^{+}\right]}_{tot}-{\left[H^{+}\right]}_{1}}{{\left[H^{+}\right]}_{2}-{\left[H^{+}\right]}_{tot}}$$

Based on the known pH value in the medium, which is usually one of the parameters measured during the process of cell cultivation, an algorithm can be created to automate the input flows in order to achieve a constant pH level on the sensor.

### Comsol Simulations

Performances of the proposed mixer are analyzed using the Comsol Multiphysics^®^ 6.0^®^ software tool (Stockholm, Sweden). The software solves different partial differential equations by using Finite Element Method for the structure and proper physics interface. For the micromixing simulations, physics interfaces Laminar flow and the Transport of Diluted species are used. Laminar flow describes the physics of fluid flow in microchannels while Transport of Diluted species enables the mixing of liquids with different color concentrations for the design presented in Figure 2a. In the used model, fluid properties have been considered, such as water with a density of 1000 $\frac{\mathrm{kg}}{m^{3}}$, and dynamic viscosity 8.9 × $10^{-4}$ Pa·s with no-slip boundary conditions.

To investigate the influence of liquid mixing, simulations of two liquids with concentrations of 0 and 1 mol/m^3^ have been performed for different flow rates. As a measure of homogeneity of the mixed liquids along the cross-section of defined Probes at the inlet, first, second, third, and last serpentine, and outlet (Figure 2a); performances of the mixers are described with a mixing index (MI). The MI takes value between 0 and 1, for completely unmixed and mixed liquids, respectively. Equation (4) presents MI along the channel cross-section:(4)$$\mathrm{MI}=1-\frac{\int \left|c-c_{\infty }\right|\mathrm{dx}}{\int \left|c_{0}-c_{\infty }\right|\mathrm{dx}}$$
where c presents the normalized concentration along the channel cross-section, $c_{\infty }\;$is a normalized concentration for well-mixed liquids (in our case 0.5) and$\;c_{0}$ normalized concentration of initial mixing of liquids (in our case 1). If the mixing index takes a value higher than 0.9, then one can consider that liquids are perfectly mixed and good mixing performances will be considered if the MI has a value higher than 0.8.

In the proposed LOC, the first micromixer is used to mix acetic acid (pH 4.5) and a cell medium (pH 7.4) to adjust the pH, while the second one is used to mix an enzyme and the mixed liquid from the first. Considering Equation (3) and the ratio of flow rates at inlets of the first micromixer, analysis of the mixing properties is performed for the flow rate ratio equal to 0.33. Results of the MI are presented for simulated values of flow rates in the range of 10 to 150 µL/min for different cross-sections (Probes) in the structures, Figure 2b. In general, in the case of low values of flow rate, the dominant mechanism of mixing is the diffusion process. On the other hand, for higher values of flow rate, the Reynolds number has a value greater than 1, which means that lateral convection becomes increasingly important compared to diffusion. Consequently, the MI value decreases. However, considering that proposed mixers have 35 repeating units, the proper mixing is achieved even for high values of flow rates.

In the second micromixer, the goal is to mix the enzyme and medium in the same volume ratio. To achieve this, the flow rate of enzyme entering the second micromixer must be equal to the sum of flow rates of the acetic acid and medium entering the first micromixer. In other words, the sum of the flow rates at the inlets of the first micromixer should be equal to the flow rate at the inlet of the second micromixer. For that reason, Figure 2c presents validation results for the flow rate equal to 67 µL/min applied at both inlets, which is corresponding to the sum for the flow rate equal to $Q_{1}=50\;µL$/min and $Q_{2}=16.7\;µL$/min. The results in Figure 2c demonstrate good matching between simulation results and mixing experiments.

## 3. Materials and Methods

### 3.1. Materials

#### 3.1.1. Materials for LOC Fabrication

The proposed LOC has been realized as a multilayer structure, with a 3D printed middle layer realized using SLA 3D printing technology (low-force stereolithography, LF-SLA), on a Formlabs Form3 3D printer. The flexible transparent resin with 80A shore hardness (Flexible 80A) from Formlabs has been used. The microfluidic chip is closed from the top and bottom side with transparent PMMA layers cut via CO_2_ laser CNC—MBL 4040RS (Minoan Binding Laminating, Belgrade, Serbia).

#### 3.1.2. Chemicals

Initially, DropSens 220 AT electrodes (Metrohm) have been tested with different concentrations of Glc (D-(+)-Glucose, ACS reagent, Sigma-Aldrich, St. Louis, MO, USA) prepared in 50 mM acetate buffer (pH 5.1), Sigma-Aldrich. Diluted acetic acid (Acetic Acid ACS Reagent >=99.7%, Sigma-Aldrich) with pH 4.5 is used for pH adjustment. After mixing enzyme (Glucose Oxidase from *Aspergillus Niger*, Type X-S, Sigma-Aldrich) with different concentrations of Glc, the electrochemical response has been measured via cyclic voltammetry (CV) and chronoamperometry, measured with Potentiostat/Galvanostat/Impedance analyzer PalmSens4 (PalmSens BV, Houten, The Netherlands) and PSTrace 5.8 software. As a non-specific sample for the enzyme, fructose (D(-)Fructose, Sigma-Aldrich) has been used, prepared as 5 mg/mL in acetate buffer. Finally, cell medium (Dulbecco’s Modified Eagle’s Medium-high Glucose, w/ 4500 mg/L Glucose, L-glutamine, sodium pyruvate, and sodium bicarbonate, liquid, sterile-filtered, suitable for cell culture, Sigma-Aldrich) has been used for real-sample detection in the proposed LOC.

### 3.2. Methods

#### 3.2.1. Fabrication and Characterization of the LOC

The layout of the proposed LOC with an integrated DropSens electrode is presented in Figure 3. The MF chip has been designed in hybrid technology, utilizing PMMA slabs that sandwiched 3D printed MF channels. Microfluidic channel has been fabricated using SLA 3D printing technology. The flexible transparent resin with 80A shore hardness has been used for the fabrication of the MF channels layer. This material has been chosen since, after printing, it has appropriate flexibility and, therefore, can be easily sandwiched between two PMMA plates. Namely, the top of the printed layer with open channels is adhering to PMMA easily and a small force is needed to keep the channels confined. It should be noted that during the tests of the proposed MF chip fabricated in Flexible 80A resin closed with two PMMA plates the flow rates of 15 mL/min without leakage can be achieved.

The proposed resin and utilized 3D printing technology enabled the production of a 3D printed layer with open channels of 200 µm × 200 µm. The layer height during printing is set to 100 µm, while other printing parameters have been chosen for the Flexible 80A resin according to manufacturer recommendations [56]. The model has been printed directly on the build plate with open channels pointing upwards to overcome any demand for support structures. After this, printing channels have been thoroughly washed with isopropyl alcohol (IPA, Sigma-Aldrich) and left in the IPA for 20 min to soak. Afterwards, the MF channels have been washed with a syringe-filled IPA under pressure to remove any left residue resin inside the channels. After washing, the printed MF channels have been left to fully dry overnight, followed by a curing process of two minutes exposing them to 365 nm and 400 nm UV light (Anycubic Wash and Cure v1).

The serpentine-based MF mixers are realized via SLA printing in Layer 2, containing three inlets for the medium, acetic acid, and enzyme. At the end of the second serpentine mixer, the chamber for the mixed liquid is placed, which is positioned at the top of the sensing part of the DropSens electrode. Layer 2, together with the DropSens electrode, is sandwiched between Layers 1 and 3, which have been realized by micromachining of 2- and 6-mm-thick PMMA layers, respectively. In addition, Layer 3 contains a place for the electrode which is realized by scanning the surface with CO_2_ laser (parameters: power: 35 W, speed 15 mm/s) for the electrode integration in order to make a slot. The 3D profile of the realized microfluidic channel shows well-defined edges and uniform channel walls, Figure 3b, while Figure 3c presents a realized chip with an integrated electrode. The overall chip dimensions are 69.25 mm × 46.65 mm, with an inlet radius of 1 mm, radius of screws 1.5 mm, and the chamber radius of 4 mm. In addition, the mixer’s length is 38.4 mm while its width is 8.2 mm.

#### 3.2.2. Electrochemical Measurements

In all electrochemical experiments, screen-printed DropSens 220AT electrodes are used which contain gold working and counter electrode, while reference electrode is made of silver/silver chloride (Ag/AgCl). Electrochemical cleaning via CV (potential range −0.2 V to 1.5 V vs. Ag/AgCl reference electrode, scan rate 0.5 V/s, 60 scans) has been conducted for all DropSens electrodes. Afterward, electrodes have been rinsed with MiliQ water and integrated into the MF chip. Glucose sensing principle is based on electrochemical detection of H_2_O_2_, which is the product of reaction of Glc and GOx. In order to find the potential of oxidation of H_2_O_2_, CV (potential range −0.2 V to 1.5 V vs. Ag/AgCl reference electrode, 8 scans) has been performed with the low scan rate 0.05 V/s in order to enable manual adding of additional droplets at the electrode surface. In that way, real-time detection of reaction can be followed in CV. Afterward, chronoamperometry was used for measurements of different concentrations of Glc during 60 s.at the fixed potential of H_2_O_2_ oxidation, 0.9 V vs. Ag/AgCl reference electrode. Solutions of GOx, Glc, and fructose have been prepared in 50 mM acetate buffer (pH 5.1).

## 4. Results and Discussion

In the proposed LOC with integrated sensor, the Glc molecules are oxidized in the chemical reaction with the GOx along the microfluidic flow. After complete mixing with the medium of interest, and a fixed amount of time, the side-product of the reaction, H_2_O_2_, is oxidized and produces the electrochemical signal at the certain voltage. The used screen-printed 3-electrode system integrated inside the MF chip with a golden working/counter electrode and Ag/AgCl reference electrode demonstrates a H_2_O_2_ oxidation peak at around 0.9 V.

Figure 4a presents the results of the CV experiment where the specificity and potential of H_2_O_2_ oxidation are examined. In this experiment, after every two scans additional droplet of 40 µL of analyte is added. The first two scans in the cyclic voltammogram, presented in Figure 4a contain the response of acetate buffer. Scans 3 and 4 present CV response after adding a droplet of 1 mg/mL GOx, and scans 5 and 6 present CV response after adding a droplet of 5 mg/mL fructose. The intensity of oxidation peaks at 0.9 V is shown in Figure 4b. The results show that the mixed fructose and enzyme did not give a response, which is expected since GOx is a non-specific enzyme for fructose. In addition, scans 5 and 6 show that after adding a droplet of Glc, a typical fingerprint of H_2_O_2_ in a voltammogram is produced, resulting in an increase in current.

The calibration of the electrochemical sensor is performed in an acetate buffer. The results of chronoamperometry for different concentrations of Glc mixed in acetate buffer over time are proposed in Figure 4c. The Glc concentration has been varied in the range from 0.1 mg/mL to 50 mg/mL. The calibration curve of Glc detection in acetate buffer is shown in Figure 4d, where the signal for different Glc concentrations is presented as a relative change versus acetate buffer response measured after 60 s of reaction. It can be seen that the H_2_O_2_ increases with the increasing concentration of Glc resulting in an exponential increase of output current. The region of interest for the detection of Glc in a cell medium is presented in the inset of Figure 4d.

Considering that the Glc content is one the most crucial variables in the cell culturing process, the concentration of the Glc is also measured in cell culturing media. Results of chronoamperometry for different concentrations of Glc in cell medium for Dulbecco’s Modified Eagle’s Medium with the Glc concentration of 4.5 mg/mL are shown in Figure 5a. The medium was diluted to detect different concentrations of glucose, while keeping the pH value of the medium constant. Since the pH of the medium is 7.4, its pH value is first adjusted to 5.1 by adjusting the corresponding flow rates to 0.33 at the inlet of the first mixer, as has been presented in the simulation results (Figure 2b). This resulted in a decrease in the total concentration of Glc in the sample, but results in a faster reaction time with the enzyme. The measurement results after mixing with the enzyme and detection are shown in Figure 5a for different concentrations of Glc, including dilutions with acetic acid and enzyme. The signal in the medium is lower compared to pure acetate buffer (inset in Figure 4d) due to the presence of other ions in the medium, such as L-glutamine, sodium pyruvate, sodium bicarbonate, etc. For practical applications, it is important to recalculate the initial concentration of Glc in the macrosystem, before entering the LOC, diluting with acetic acid, and reacting with the enzyme. The calibration curve of Glc detection in the cell medium is shown in Figure 5b, where the sensor’s response is presented in the function of real Glc concentration, before entering the LOC. The signal is presented as a relative change versus acetate buffer. Finally, it can be seen that the proposed system allows real-time Glc measurement in liquid analyte with the linear change in the current with the Glc concentration.

The proposed concept has several advantages for potential use in different biomedical applications. Namely, glucose concentration in the blood is the most widely used measure for monitoring blood sugar levels. Normal blood glucose levels are between 0.7 and 1 mg/mL [57], and anything higher than that could indicate diabetes or another underlying medical condition. Saliva also contains glucose, but the concentration is much lower compared to blood, usually ranging from 5 × 10^−4^ to 10^−3^ mg/mL [58]. The glucose concentration in saliva is used to monitor the changes in blood sugar levels, and it is also used in some diagnostic tests. The concentration of glucose in urine varies based on blood glucose levels, with a range of 0–0.20 mg/mL or more, depending on the blood glucose levels [59]. The proposed concept has been able to detect previously mentioned concentration ranges; therefore, it can be used for applications with these liquid samples.

In addition, due to the flow system design, there is no need of immobilizing the enzyme at the sensor surface of the gold electrode together with a kind of solid mediator, which is always essential since the washout problem arises after some time of operation. Recently, various solutions have been proposed for glucose sensing based on a LOC system that incorporates micromixers. These solutions utilize spectroscopic [60,61], colorimetric [62], or electrochemical methods for the detection of Glc [63]. However, despite these advances, none of these proposed systems have been tested in real samples, and none of them have been designed for seamless integration with actual systems. The proposed concept for sensing can acquire samples automatically without the need for manual sampling and can allow measurement in real time. As an in-line sensor, the proposed chip can allow monitoring of the nutrient in the cell medium while, at the same time, preventing potential contamination of the medium during the sampling process. In addition, the proposed chip allows adjusting the reaction according to the pH value, so it is a particularly interesting application for in-line measurements in the cell cultivation process where the glucose and pH change over time.

## 5. Conclusions

In this paper, a novel 3D printed microfluidic chip with integrated serpentine-based micromixers and an electrochemical sensor for glucose sensing in a liquid analyte was proposed. Numerical simulations for optimizing the mixing performances were performed and a theoretical model for fluid mixing was utilized for adjusting the inputs fluid flows according to the pH values of the sample. The sensing potential was demonstrated for Glc concentrations measurement in acetate buffer and cell culture media. The proposed chip was characterized by the usage of small volumes of the sample and reagents, good sensitivity, and linear response. Therefore, it showed great potential for direct in-line monitoring in macrosystems such as bioreactors where the glucose and pH change over time. The proposed platform has a high potential for different applications in the processes where the microsystem can enable monitoring of the relevant parameters for the macrosystem such as bioreactors.

## Figures and Tables

**Figure 1 micromachines-14-00503-f001:**
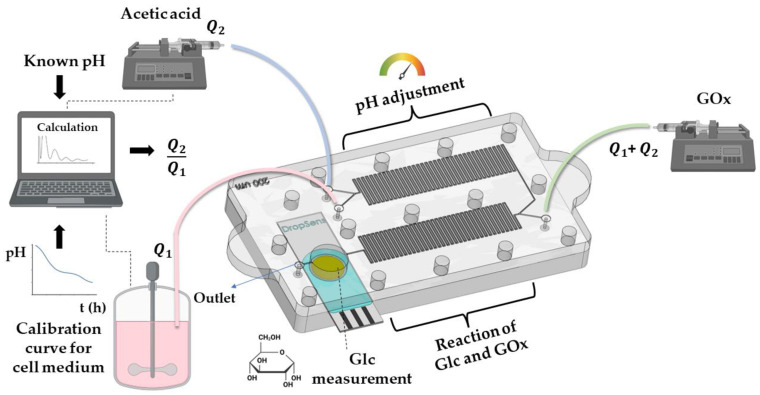
Schematics of the proposed glucose detection concept.

**Figure 2 micromachines-14-00503-f002:**
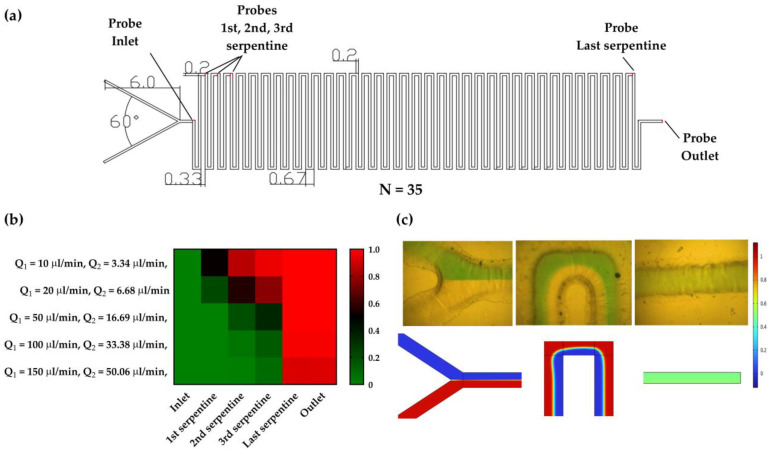
(**a**) Design of serpentine-based micromixer with Probes used in simulations. Dimensions are in mm. (**b**) Mixing index values for different flow rates with ratio 0.33. (**c**) Comparison between simulation results and mixing of color dyes in the microfluidic mixer for flow rate equal to 67 µL/min at inlets, after first serpentine and at outlet.

**Figure 3 micromachines-14-00503-f003:**
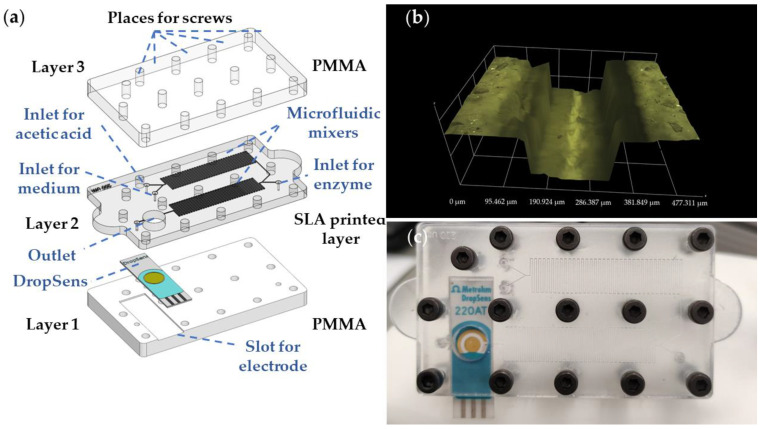
(**a**) Multilayer structure of the proposed LOC. (**b**) Three-dimensional profile of the SLA printed MF channel. (**c**) Realized LOC with integrated DropSens sensor.

**Figure 4 micromachines-14-00503-f004:**
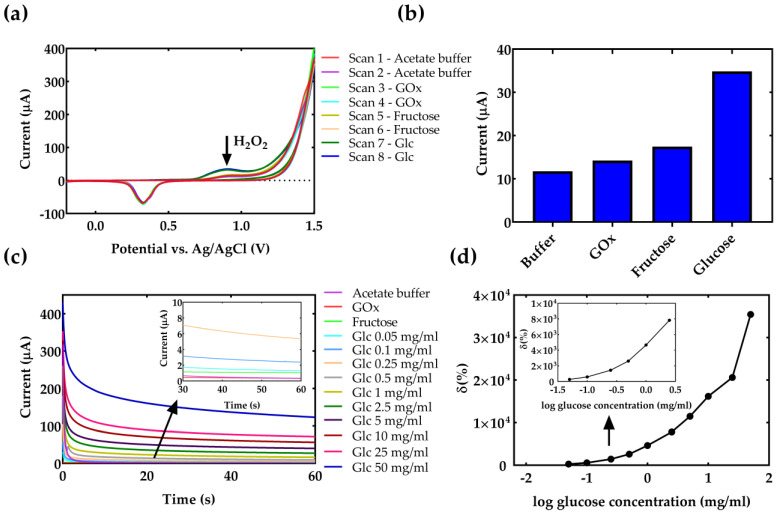
Results of electrochemical Glc detection in the proposed LOC. (**a**) Examining the potential of H_2_O_2_ oxidation peak. Scans 1–2 in acetate buffer. Scans 3–4 added droplet of GOx. Scans 5–6 added droplet of fructose. Scans 7–8 added droplet of Glc. (**b**) Intensity of oxidation peaks from Figure 4a at 0.9 V vs. Ag/AgCl. (**c**) Results of chronoamperometry for Glc prepared in acetate buffer. (**d**) Calibration curve of Glc detection in acetate buffer. The signal is presented as a relative change versus acetate buffer response. $\delta \left(\%\right)=100\times \left(I_{cc}-I_{buffer}\right)/I_{buffer}$.

**Figure 5 micromachines-14-00503-f005:**
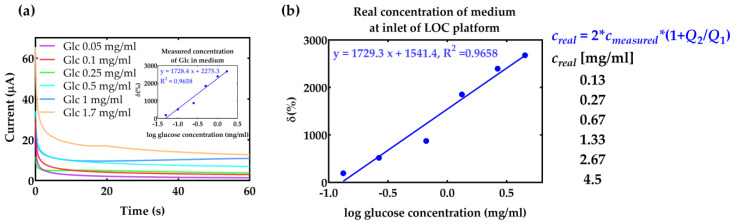
(**a**) Results of chronoamperometry for different concentrations of Glc in cell medium Inset: Calibration curve of Glc detection for medium diluted with acetic acid and enzyme. (**b**) Calibration curve of Glc detection in the cell medium for Glc concentrations at inlet of LOC. The real concentration is calculated by multiplying the measured concentration with the flow rate factor and factor 2 which presents dilution with enzyme mixing. The signal is presented as a relative change versus acetate buffer response: $\delta \left(\%\right)=100\times \left(I_{cc}-I_{buffer}\right)/I_{buffer}$.

## Data Availability

The data presented in this study are available on request from the corresponding author.

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
