# Peer review of "3D-Printed Microfluidic Chip for Real-Time Glucose Monitoring in Liquid Analytes"

_micromachines, 2023, doi:10.3390/mi14030503_

Round 1

Reviewer 1 Report

The paper presents an interesting sensor for glucose monitoring.

But I do not agree with the designation “platform” for it. The microfluidic platform is a usually more than a sensor. (see the definition given by Daniel Mark in 2009 in the paper Microfluidic lab-on-a-chip platforms: requirements, characteristics and applications).

Since the paper uses it quite often, it will be too much effort to replace it. But please remember it for the next paper.

Some more remarks:

In the paragraph explaining the fabrication technologies (line 82) the mSLA (masked stereo lithography) is missing.

In the paragraph explaining possible applications of the sensor (lines 83-100): please specify the needed range of measurement of glucose in the different applications in order to see which applications are target applications for the proposed sensor (besides of the cell culture).

In the simulation paragraph:

Please specify the details of the simulation model (e.g. number and size of the cells, discretization and stabilization methods – when using laminar flow in combination with transport of diluted species this becomes important).

In line 234:

Please specify the curing apparatus.

In the results and discussion section:

Can You please specify the timing: Which is the sample rate You can measure? Is there a cleaning procedure needed between different measurements?

What about reproducibility and limit of detection? (These parameters are important when the sensor is more than a proof of concept.) If You do not have the measurements yet, mention in the outlook that these will be investigated soon.

Author Response

Dear Sir/Madam,

We appreciate the careful reviewing of our manuscript and we gratefully acknowledge the Reviewers and Editor for their useful comments. We have modified the manuscript to answer the questions raised by the Reviewers. Our answers, some additional comments, and explanations are given in the text below and the changes are made in the resubmitted manuscript.

We thank you for the opportunity to improve our paper, including your comments and suggestions.

Sincerely,

The authors

Answers to the Reviewer’s Comments

The paper presents an interesting sensor for glucose monitoring.

Our reply: We appreciate the careful reviewing of our manuscript and we gratefully acknowledge the Reviewer. Our answers and additional explanations are given below.

Comment 1: I do not agree with the designation “platform” for it. The microfluidic platform is usually more than a sensor. (see the definition given by Daniel Mark in 2009 in the paper Microfluidic lab-on-a-chip platforms: requirements, characteristics and applications). Since the paper uses it quite often, it will be too much effort to replace it. But please remember it for the next paper.

Our reply: We would like to thank the Reviewer to point that out. Instead of “platform” we use the word “chip” in the resubmitted manuscript.

Comment 2: In the paragraph explaining possible applications of the sensor (lines 83-100): please specify the needed range of measurement of glucose in the different applications in order to see which applications are target applications for the proposed sensor (besides of the cell culture).

Our reply: The authors are grateful for the Reviewer’s comment. An additional explanation is included in the section Results and Discussion since in that section we included the potential biomedical application of the proposed platform.

Lines 336-346

The proposed concept has several advantages for potential use in different bio-medical applications. Namely, glucose concentration in the blood is the most widely used measure for monitoring blood sugar levels. Normal blood glucose levels are between 0.7 - 1 mg/ml [57], and anything higher than that could indicate diabetes or another underlying medical condition. Saliva also contains glucose, but the concentration is much lower compared to blood, usually ranging from 5 ×10-4 – 10-3 mg/ml [58]. The glucose concentration in saliva is used to monitor the changes in blood sugar levels, and it is also used in some diagnostic tests. The concentration of glucose in urine varies based on blood glucose levels, with a range of 0 - 0.20 mg/ml or more, depending on the blood glucose levels [59]. The proposed concept has been able to detect previously mentioned concentration ranges, and since that it can be used for applications with these liquid samples.

Comment 3: In the simulation paragraph: Please specify the details of the simulation model (e.g. number and size of the cells, discretization and stabilization methods – when using laminar flow in combination with the transport of diluted species this becomes important).

Our reply: Considering the Reviewer's valuable remark, the authors would like to give additional information about the simulation model.

Number and size of the cells: 551450 elements, minimum element quality: 0.0184, average element quality: 0.9951, element area ratio: 0.002681;

Discretization: Concentration – Cubic;

Stabilization methods: Consistent Stabilization: Streamline and Crosswind diffusion, Crosswind diffusion type: Codina, Equation residual: Full residual.

Comment 4: In line 234:

Please specify the curing apparatus.

Our reply: Authors are grateful for the Reviewer's observation. We included the model of curring apparatus in the Materials and Methods section.

Lines 238-240

After washing, the printed MF channels have been left to fully dry overnight, followed by a curing process of two minutes exposing them to 365 nm and 400 nm UV light (Anycubic Wash and Cure v1).

Comment 5: In the results and discussion section: Can You please specify the timing: Which is the sample rate You can measure? Is there a cleaning procedure needed between different measurements?

Our reply: The authors would like to give an additional explanation for the research design.

In practical applications, if the glucose increases over time, the system can work in continuous mode, where the relative change of glucose can be detected with the sensor.

In the case of a cellular medium, where glucose and pH are changing during the cultivation process, periodic maintenance of the sensor is required. In the proposed configuration the sensor has to be replaced since the saturation of the surface can occur, which can block detection of lower values of the signal. Therefore, in the proposed configuration low-cost sensor is integrated as a USB slot and can be easily changed between measurements.

Changes in glucose and pH parameters during cultivation depend on the size of the bioreactor and the amount of medium. However, in general, this change is slow and sampling is done at a level of several hours or more.

Comment 6: What about reproducibility and limit of detection? (These parameters are important when the sensor is more than a proof of concept.) If You do not have the measurements yet, mention in the outlook that these will be investigated soon.

Our reply: We thank the Reviewer for the comment, we are providing an additional explanation for the Reviewer’s question.

The detection limit in this configuration is determined by the sensor performance, which can be improved for a different application by changing the sensor geometry or using nanomaterials to enhance sensitivity. Since in this study, commercial screen-printed electrodes have been used, the quality of the sensor response depends on the electrodes’ production quality. However, reproducibility in this study has been ensured by using the relative change of the signal versus the acetate buffer for each electrode. In other words, even if electrodes are ideally the same, i.e., the signal intensity is different at two electrodes due to the surface porosity, paste quality, etc. their relative change versus acetate buffer eliminates all differences between different electrodes.

Reviewer 2 Report

This paper presents.

In this paper, a 3D-printed microfluidic (MF) platform for glucose (Glc) sensing in a liquid analyte is proposed. The platform contains integrated serpentine-based micromixers with a slot for USB-like integration of commercial DropSens electrodes. After adjusting the sample’s pH in the first micromixer, small volumes of the sample and enzyme are mixed in the second micromixer and lead to a sensing chamber where the Glc concentration is measured via chronoamperometry. The sensing potential was examined for Glc concentrations in acetate buffer in the range of 0.1 - 100 mg/ml and afterward tested for Glc sensing in a cell culturing medium. The proposed platform showed great potential for connection with macrosystems like bioreactors, for direct in-line monitoring of a quality parameter in a liquid sample.

1.     Abstract is directly started from the current study detail. Where is the application and importance of this topic with gap or novelty?

2.     In figure 2-part c shows that the pattern of flow is not exactly matching any reason.

3.     Why channels are so long with too many units for mixing?

4.     Simulation results for flow are missing

5.     Physics and boundary conditions?

6.     Mesh independence?

Author Response

Dear Sir/Madam,

We appreciate the careful reviewing of our manuscript and we gratefully acknowledge the Reviewers and Editor for their useful comments. We have modified the manuscript to answer the questions raised by the Reviewers. Our answers, some additional comments, and explanations are given in the text below and the changes are made in the resubmitted manuscript.

We thank you for the opportunity to improve our paper, including your comments and suggestions.

Sincerely,

The authors

Answers to the Reviewer’s Comments

This paper presents.

In this paper, a 3D-printed microfluidic (MF) platform for glucose (Glc) sensing in a liquid analyte is proposed. The platform contains integrated serpentine-based micromixers with a slot for USB-like integration of commercial DropSens electrodes. After adjusting the sample’s pH in the first micromixer, small volumes of the sample and enzyme are mixed in the second micromixer and lead to a sensing chamber where the Glc concentration is measured via chronoamperometry. The sensing potential was examined for Glc concentrations in acetate buffer in the range of 0.1 - 100 mg/ml and afterward tested for Glc sensing in a cell culturing medium. The proposed platform showed great potential for connection with macrosystems like bioreactors, for direct in-line monitoring of a quality parameter in a liquid sample.

However, I indicated the following elements to revision:

Our reply: The authors acknowledge the Reviewer's detailed review of the paper. Responses and additional explanations are given in below.

Comment 1: Abstract is directly started from the current study detail. Where is the application and importance of this topic with gap or novelty?

Our reply: We thank the Reviewer for the valuable comment. Additional sentences are included in the paper's abstract.

Lines 14-19

The connection of macrosystems with microsystems for in-line measurements is important in different biotechnological processes as it enables precise and accurate monitoring of process parameters at a small scale, which can provide valuable insights into the process and ultimately lead to improved process control and optimization. Additionally, it allows for continuous monitoring without the need for manual sampling and analysis, leading to more efficient and cost-effective production. In this paper, a 3D-printed microfluidic (MF) chip for glucose (Glc) sensing in a liquid analyte is proposed. The chip contains integrated serpentine-based micromixers with a slot for USB-like integration of commercial DropSens electrodes. After adjusting the sample’s pH in the first micromixer, small volumes of the sample and enzyme are mixed in the second micromixer and lead to a sensing chamber where the Glc concentration is measured via chronoamperometry. The sensing potential was examined for Glc concentrations in acetate buffer in the range of 0.1 - 100 mg/ml and afterward tested for Glc sensing in a cell culturing medium. The proposed chip showed great potential for connection with macrosystems like bioreactors, for direct in-line monitoring of a quality parameter in a liquid sample.

Comment 2: In figure 2-part c shows that the pattern of flow is not exactly matching any reason.

Our reply: The authors agree with Reviewer’s observation. An additional explanation is given below.

Due to the imperfections of the fabrication process, it is impossible to create a meandered-based structure with sharp edges. In the design setup for 3D printing, rounded edges were used because sharp edges lead to clogging of the channels. There is a small difference between simulated and measured values that do not impact the evaluated performance of the mixer.

Results in Figure 2c present two liquids with different color concentrations that have a parallel flow in the microfluidic channel, due to the laminar behavior of liquids at the microscale. At the interface of two liquids, there is a tin layer of mixed liquids due to the diffusion process that occurs between two liquids. That layer can be seen in experimental results as a shadow between yellow and blue colors. Unfortunately, the image quality depends on the used microscope performance. The limited transparency of the chip and its thickness affect the focusing of the microscope. and authors cannot provide a better image of the experiment.

Comment 3: Why channels are so long with too many units for mixing?

Our reply: The authors appreciate Reviewer's valuable remark.

Considering that the proposed application of the microfluidic chip is for real-time monitoring of glucose in the bioreactor system, it is important to enable good mixing performance even for high values of the flow rate. In addition, with a higher number of repeating units in microfluidic mixers, besides good mixing of two liquids, a reaction time for glucose and enzyme is provided, since the reaction time dictates the production of H2O2 which has been used for electrochemical detection.

Comment 4: Simulation results for flow are missing

Our reply: The authors appreciate Reviewer's question. However, simulation results for the flow are presented in Figures 2b and 2c. Figure 2b through the heat map contains the calculated mixing index from simulation results at different probes, placed over the mixer's length. In addition, color concentration comparison with real experiments is presented in Figure 2c.

Comment 5: Physics and boundary conditions?

Our reply: The authors agree with the Reviewer's suggestion. Additional information is included in the resubmitted manuscript.

For the micromixing simulations, physics interfaces Laminar flow and the Transport of Diluted species are used. Laminar flow describes the physics of fluid flow in microchannels while Transport of Diluted species enables the mixing of liquids with different color concentrations for the design presented in Figure 2a. In the used model fluid properties have been considered like water with a density of 1000 , and dynamic viscosity 8.9 × 10-4 Pa s with no-slip boundary conditions.

Comment 6: Mesh independence?

Our reply: We thank the Reviewer for the detailed analysis of the results. In order to discuss the Reviewer's question, additional simulations were performed and the results and analysis are presented below. Additional information about the user-controlled mesh is described.

The user-controlled mesh is set in the model by mapping the micromixers' surface (option Mapped mesh) with lines parallel to the channels' edges (option Distribution). In that way, the mesh is set with elements made of squares and rectangles, which can properly describe the interface between two liquids moving in parallel layers through the microchannels.

The results in Figure below present mixing index results for the probe at 3rd serpentine for the number of mesh elements in the range 9 × 104 to 5.5 × 105. The results show that the mixing index parameter has the same value for the increased number of mesh elements, and thus, presents the mesh independence of the results.

Reviewer 3 Report

The authors report a 3D-printed microfluidic platform to detect glucose in a liquid analyte. This platform includes serpentine-based micromixers with a slot for USB-like integration of commercial electrodes. This manuscript can be enhanced based on the following topics:
1.- The abstract should add more information on the materials and main results.
2.- The introduction could present the main advantages of the proposed microfluidic platform compared to others reported in the literature.
3.- The authors should consider more information on the different components of the microfluidic platform. In addition, more information on the materials and dimensions of the different components of the microfluidic platform can be incorporated.
4.- The resolution of Figures 2 and 3 could be improved.
5.- The fabrication process of the microfluidic platform should include more detailed information.
6.- The discussion of the main results could be improved.
7- What are the main limitations of the proposed microfluidic platform?
8.-What are the future research works?
9.- The conclusions could be enhanced by considering the above comments.

Author Response

Dear Sir/Madam,

We appreciate the careful reviewing of our manuscript and we gratefully acknowledge the Reviewers and Editor for their useful comments. We have modified the manuscript to answer the questions raised by the Reviewers. Our answers, some additional comments, and explanations are given in the text below and the changes are made in the resubmitted manuscript.

We thank you for the opportunity to improve our paper, including your comments and suggestions.

Sincerely,

The authors

Answers to the Reviewer’s Comments

The authors report a 3D-printed microfluidic platform to detect glucose in a liquid analyte. This platform includes serpentine-based micromixers with a slot for USB-like integration of commercial electrodes. This manuscript can be enhanced based on the following topics:

Our reply: The authors are grateful to the Reviewer for the detailed analysis and review of the paper.

Comment 1: The abstract should add more information on the materials and main results.

Our reply: The authors are grateful for the Reviewer’s valuable suggestion. Additional information is added in the abstract.

The connection of macrosystems with microsystems for in-line measurements is important in different biotechnological processes as it enables precise and accurate monitoring of process parameters at a small scale, which can provide valuable insights into the process and ultimately lead to improved process control and optimization. Additionally, it allows continuous monitoring without the need for manual sampling and analysis, leading to more efficient and cost-effective production. In this paper, a 3D-printed microfluidic (MF) chip for glucose (Glc) sensing in a liquid analyte is proposed. The chip made in Poly(methyl methacrylate) - PMMA contains integrated serpentine-based micromixers realized via stereolithography with a slot for USB-like integration of commercial DropSens electrodes. After adjusting the sample’s pH in the first micromixer, small volumes of the sample and enzyme are mixed in the second micromixer and lead to a sensing chamber where the Glc concentration is measured via chronoamperometry. The sensing potential was examined for Glc concentrations in acetate buffer in the range of 0.1 - 100 mg/ml and afterward tested for Glc sensing in a cell culturing medium. The proposed chip showed great potential for connection with macrosystems like bioreactors, for direct in-line monitoring of a quality parameter in a liquid sample.

Comment 2: The introduction could present the main advantages of the proposed microfluidic platform compared to others reported in the literature.

Our reply: The authors thank the Reviewer for the suggestion. Additional sentences are added in the revised manuscript in the section Discussion according to the Reviewer's remark.

Line 350-354

Recently, various solutions have been proposed for glucose sensing based on a LOC system that incorporates micromixers. These solutions utilize spectroscopic [60,61], colorimetric [62], or electrochemical methods for the detection of Glc [63]. However, despite these advances, none of these proposed systems have been tested in real samples, and none of them have been designed for seamless integration with actual systems.

Comment 3: The authors should consider more information on the different components of the microfluidic platform. In addition, more information on the materials and dimensions of the different components of the microfluidic platform can be incorporated.

Our reply: We thank the Reviewer for this suggestion. Additional information about materials and dimensions is added to the section Materials and Methods.

Lines 241 - 253

The serpentine-based MF mixers are realized via SLA printing in Layer 2containingng three inlets for the medium, acetic acid, and enzyme. At the end of the second serpentine mixer, the chamber for the mixed liquid is placed, which is positioned at the top of the sensing part of the DropSens electrode. Layer 2 together with the DropSens electrode is sandwiched between Layers 1 and 3, which have been realized by micromachining of 2 and 6-mm-thick PMMA layers, respectively. In addition, Layer 3 contains a place for the electrode which is realized by scanning the surface with a CO2 laser (parameters: power: 35 W, speed 15 mm/s) for the electrode integration in order to make a slot. The 3D profile of the realized channel shows well-defined edges and uniform channel walls, Figure 3b, while Figure 3c presents a realized chip with an integrated electrode. The overall chip dimensions are 69.25 mm × 46.65 mm, with an inlet radius of 1 mm, radius of screws 1.5 mm, and the chamber radius of 4 mm. In addition, the mixer's length is 38.4 mm while its width is 8.2 mm.

Comment 4: The resolution of Figures 2 and 3 could be improved.

Our reply: The authors are grateful to the Reviewer for this comment. Figures 2 and 3 are changed with Figures with better resolution.

Comment 5: The fabrication process of the microfluidic platform should include more detailed information.

Our reply: We add more detailed information about the fabrication process in Lines 241-253.

Comment 6: The discussion of the main results could be improved.

Our reply: The authors are grateful for the Reviewer's comment. We improved the discussion section with the following lines.

Lines 336-361

The proposed concept has several advantages for potential use in different bio-medical applications. Namely, glucose concentration in the blood is the most widely used measure for monitoring blood sugar levels. Normal blood glucose levels are between 0.7 - 1 mg/ml [57], and anything higher than that could indicate diabetes or another underlying medical condition. Saliva also contains glucose, but the concentration is much lower compared to blood, usually ranging from 5 ×10-4 – 10-3 mg/ml [58]. The glucose concentration in saliva is used to monitor the changes in blood sugar levels, and it is also used in some diagnostic tests. The concentration of glucose in urine varies based on blood glucose levels, with a range of 0 - 0.20 mg/ml or more, depending on the blood glucose levels [59]. The proposed concept has been able to detect previously mentioned concentration ranges, and since that it can be used for applications with these liquid samples.

In addition, due to the flow system design, there is no need of immobilizing the enzyme at the sensor surface of the gold electrode together with a kind of solid mediator, which is always essential since the washout problem arises after some time of operation. Recently, various solutions have been proposed for glucose sensing based on a lab-on-a-chip system that incorporates micromixers. These solutions utilize spectroscopic [60,61], colorimetric [62], or electrochemical methods for the detection of Glc [63]. However, despite these advances, none of these proposed systems have been tested in real samples, and none of them have been designed for seamless integration with actual systems. The proposed concept for sensing can acquire samples automatically without the need for manual sampling and can allow measurement in real time. As an in-line sensor proposed chip can allow monitoring of the nutrient in the cell medium while at the same time preventing potential contamination of the medium during the sampling process. In addition, the proposed chip allows adjusting the reaction according to the pH value, so it is a particularly interesting application for in-line measurements in the cell cultivation process where the glucose and pH change over time.

Comment 7: What are the main limitations of the proposed microfluidic platform?

Our reply: The authors appreciate Reviewer's question.

As it was stated in the paper, the proposed microfluidic chip has been tested for very big flow rates, up to 15 ml/min, after which the connectors started to leak. In that sense, the proposed platform did not show any drawbacks or limitations. On the other hand, for some other applications, where the sensitivity of the detection has to be enhanced, the quality of the integrated commercial electrodes has to be improved. Finally, the periodic maintenance of the sensor in terms of electrochemical cleaning in acid presents the main limitation of the proposed system, due to the fact that the integrated sensors are disposable.

Comment 8: What are the future research works?

Our reply: We are grateful to Reviewer for this question.

Future work will be based on testing the proposed LOC with a continuous flow of medium from a commercial macrobioreactor system during the cultivation of different cell lines. In addition, as it was stated in the Discussion in the paper, potential use in biomedical applications will be considered with samples like saliva, urine, blood, etc. Moreover, the proposed systems' sensitivity will be examined after the modification of commercial electrodes with advanced nanomaterials in order to improve the sensors' limit of detection and sensitivity for biomedical applications. Finally, instead of commercial electrodes, custom-made solutions of electrodes with enhanced sensitivity can be used instead of commercial DropSens electrodes.

Comment 9: The conclusions could be enhanced by considering the above comments.

Our reply: We included additional sentences in the conclusion.

Lines 363-374

In this paper, a novel 3D-printed microfluidic chip with integrated serpentine-based micromixers and an electrochemical sensor for glucose sensing in a liquid analyte was proposed. Numerical simulations for optimizing the mixing performances were performed and a theoretical model for fluid mixing was utilized for adjusting the inputs fluid flows according to the pH values of the sample. The sensing potential was demonstrated for Glc concentrations measurement in acetate buffer and cell culture media. The proposed chip was characterized by the usage of small volumes of the sample and reagents, good sensitivity, and linear response. Therefore, it showed great potential for direct in-line monitoring in macrosystems like bioreactors where the glucose and pH change over time. The proposed platform has a big potential for different applications in the processes where the microsystem can enable monitoring of the relevant parameters for the macrosystem like bioreactors.

Round 2

Reviewer 2 Report

I am satisfied with the author's response and recommend its possible publication in this journal with the addition of the following references to the existing literature.

Wearable Healthcare Monitoring Based on a Microfluidic Electrochemical Integrated Device for Sensing Glucose in Natural Sweat

Reviewer 3 Report

The authors have improved their manuscript by considering the reviewer's comments.